# Numerical Study on Mechanical Properties of the Freezing–Thawing Cycle of Tailings Based on Particle Discrete Element Method

**Gang Huang** [1,2], **Yunqin Yang** [1,2,*], **Mingyu Liu** [3], **Jianhua Zhang** [1,2], **Faping Liu** [3], **Akisa David Mwangi** [1,4] **and Haiwang Ye** [1,2]

1   School of Resources and Environmental Engineering, Wuhan University of Technology, Wuhan 430071, China; huanggang2016@whut.edu.cn (G.H.); 265012@whut.edu.cn (J.Z.); 264962@whut.edu.cn (A.D.M.); lg320855@whut.edu.cn (H.Y.)
2   Hubei Key Laboratory of Mineral Resources Processing and Environment, Wuhan 430070, China
3   Gushan Mining Co., Ltd., Anhui Masteel Mining Resources Group, Maanshan 243100, China; l13296581767@163.com (M.L.); l2098106731@163.com (F.L.)
4   Mining, Materials and Petroleum Engineering Department, Jomo Kenyatta University of Agriculture and Technology, Nairobi 62000-00200, Kenya
*   Correspondence: y907941707@whut.edu.cn

**Abstract:** To study the effects of the number of freezing–thawing cycles (F-T cycles), the dry density, and the average particle diameter on the mechanical properties of tailings, the calibration laws of the fine-scale parameters in the discrete particle element numerical simulation software PFC2D(Particle Flow Code) were first tested, and then pre-experiments were conducted in the form of orthogonal tests. Finally, according to the results of the pre-experiments and the analysis of the pre-experimental results by SPSS (Statistical Product Service Solutions) software, uniaxial tests were carried out for different numbers of freeze–thaw cycles, different dry densities, and different average particle sizes. The tailings specimens were subjected to uniaxial compression simulations. The results showed that (1) the uniaxial compressive strength of the tailings specimens decreased with each freeze–thaw cycle, and stabilized after seven freeze–thaw cycles. (2) With a greater number of freeze–thaw actions, the porosity of the tailings increased after freeze–thawing. The peak of porosity was much higher than that of the models with fewer than seven freeze–thaw actions. (3) The contact number of tailings specimens showed a significant decrease after the number of freeze–thaw cycles reached seven. However, the amount of exposure was not the main factor affecting the strength of tailings (4). As the number of freeze–thaws increased, the tailings model was more prone to stress concentration. Previously, PFC software has been applied to tailings simulation studies, and this study verifies the feasibility of this method. This research is able to offer a reference for studying the mechanical property changes of tailings in the cold highland area.

**Keywords:** tailing sand; freeze–thaw cycle; numerical simulation; mesostructured

## 1. Introduction

With the development of intelligent automation, the production efficiency of large-scale mines in China is increasing, producing more tailings. Many scholars have studied the problems of tailings disposal. At present, the most important form of tailings storage facility in the world is still the tailings pond [1].

About 70% of China's land area comprises permafrost and seasonally frozen ground, while about 91.4% of tailings piles are located in frozen-ground areas. Tailings are a discrete material with specific pores between constituent particles, and water, air, and solid particles of tailings inside the pores constitute a complex fine-scale three-phase structure. In permafrost and seasonally frozen-ground regions, F-T cycling directly affects changes in the mechanical properties of the tailings and also leads to changes in the pore pressure and

other indicators within the tailings dam body, which in turn has an impact on the stability of the tailings dam [2,3].

Moreover, tailings ponds are large-scale and prone to safety hazards. The collapse of a tailings dam can cause a serious risk to people's lives and properties downstream and in the surrounding environment. Therefore, it is necessary to investigate tailings under cyclic freeze–thaw conditions to maintain the safety and stability of tailings ponds in cold regions [1].

The current research on freeze–thaw cycling mainly comprises laboratory studies on granites and other geotechnical bodies. Many scholars have investigated the effects of freeze–thaw on the strength, anisotropy, and other properties of rocks, and found that the strength of these rocks decreases after freeze–thaw cycles. Few of these studies have dealt with tailings as a material, and few have used numerical simulations as a method of study [4–10]. Some scholars also studied the effects of freeze–thaw processes on the leaching of metals from waste rock and the oxidation of sulfides [6].

Some researchers also studied the properties of clay materials after cyclic freeze–thaw. Their work revealed that the water content, elastic modulus, and cohesion of clay decreased with the number of freeze–thaw actions; however, the internal friction angle increased [11].

Some researchers have also conducted laboratory experiments using artificial permafrost and permafrost from laboratory tests and found the effect of freeze–thaw cycle numbers is more remarkable. In contrast, the freeze–thaw temperature has a minor impact on the mechanical properties [12]. Some scholars have also studied permeability, strain rate, strain energy, temperature, surrounding pressure, water content, and soil particle size of permafrost soils [13–17].

So far, there have been many studies on tailings' mechanical properties under freeze–thaw cycles. Many researchers have also investigated water content, porosity, mineral composition, fine structure, strength, and deformation characteristics through laboratory tests [18,19] and the effect of ice lensing and freeze swelling during freeze–thaw cycles [20]. It has been suggested that temperature gradients lead to water transport during freeze–thaw processes, which changes the structure of geological bodies (sands and tailings) with discrete types [21]. Therefore, for tailings materials, the freeze–thaw cycling effect may lead to inhomogeneity of the particle structure, leading to the deterioration of mechanical properties at the macroscopic level.

For the impact of cyclic freeze–thaw on pore water transport of discrete geological bodies, the influence of freeze–thaw temperature, pore water pressure, and pore characteristics on water transport have been comparatively studied by scanning electron microscopy (SEM)-acquired fine view images. It is believed that after the cyclic freeze–thaw action, soil pores show a new alignment trend, while the pore water pressure gradually decreases. The particles become more aggregated [10]. The study of the effect of water content on the mechanical properties of soil and rock materials, such as tailings under freeze–thaw cycle conditions also has a specific basis. Some scholars have conducted impact compression experiments on granite specimens with different water contents and degrees of freezing using separated Hopkinson rods. The effect of water on the dynamic modulus of elasticity was more significant than the effect of freeze–thaw temperature [22]. Some scholars also investigated the shear strength of specimens with different water contents after freezing and thawing using fine-grained sand from the drainage field and found that the samples' shear strength decreased the most after a single freeze–thaw action, and the specimens' cohesion and internal friction angles decreased according to water content with the increasing numbers of freeze–thaw cycles.

In contrast, the cohesion of samples with the same water content was more obviously influenced by the effects of the freeze–thaw process [23]. The effect of seepage on the refined mechanics of tailings has also been studied. It was found that seepage significantly affects the particle diameter distribution of tailings particles at different depths [24].

The primary method for studying the physical and mechanical properties of tailings in the freeze–thaw zone is laboratory testing. Laboratory experiments with many freeze–thaw

cycles are difficult to realize due to the high time cost and specimen loss. The capacity of freeze–thaw equipment limits them, and the low number of specimens available for experiments and reproducibility means that it is challenging to meet the experimental requirements. The errors in the testing results are challenging to reduce. At present, many scholars have researched numerical simulation and obtained results consistent with laboratory tests [25–28], so it is feasible to conduct examinations through numerical simulation.

The first objective of the study was to explore the effects of dry density, average particle size, and number of freeze–thaw cycles on the mechanical properties of tailings. The numerical simulation of freezing–thawing cycles is a thermo-stress coupling problem, which has many precedents [29–31], but is rarely applied in the study of tailings. The second objective of this study was to verify the feasibility of this method in the study of tailings. Therefore, in this paper, we first designed orthogonal tests using SPSS software, and analyzed the numerical simulation results of pre-experiments by mathematical methods. On this basis, freeze–thaw cycles and uniaxial compression numerical simulations were carried out using PFC2D software. In accordance with the results, the mechanism and law of the changes in the mechanical properties of tailings caused by different factors were studied to provide a basis and reference for the stability study of tailings dams under the effects of freeze–thaw cycles.

## 2. Methodology

### 2.1. Pre-Experiments

An orthogonal test was designed with three factors: dry density of tailings (1480, 1640, 2240, 2460 kg/m$^3$), the average particle size after proper amplification treatment (0.632, 0.761, 0.931, 1.140 mm), and the number of freeze–thaw cycles (0, 1, 3, 7, 12, 18, 24, 30 times), and pre-experimental simulations were performed according to the orthogonal test table.

### 2.2. Numerical Simulation

The tailings with different dry densities and different average particle sizes were modeled and the model was simulated by PFC2D for different times of freeze–thaw, with the freezing temperature set to −20 °C and the thawing temperature set to 20 °C. The model was loaded after the freeze–thaw times were completed.

## 3. Numerical Simulation Mechanism and Numerical Specimen Preparation

### 3.1. Numerical Simulation Mechanism

The numerical simulation software PFC2D was used in this simulation, and the numerical simulation code used was divided into three main parts. (1) The model generation part. This part was used to generate the model and assign values to the model's modulus of elasticity, dimensions, and other parameters. (2) The thermal treatment part. In this part, the model particles were first given thermodynamic parameters, such as heat transfer coefficients, and then the model as a whole was given an initial temperature and a freeze–thaw cycle by customizing the temperature increments. This part enabled the transfer of temperature in the model, the change in temperature, and the resulting change in parameters, such as porosity. (3) The loading part. This part implemented the loading process of the model by applying velocities to the walls at the top and bottom of the model. The changes in particle displacements, stresses, and other parameters could be monitored in real time during the loading process. The initial overlap between the particles was balanced within the model by loading the surrounding pressure between the different parts to avoid errors in the final results, due to the initial contact force or initial velocity that the particles had in the subsequent run.

The model particles could apply velocity and force; the walls could only apply rate directly. Therefore, when loading the model in the loading section, if the wall was to be used to exert pressure on the particles, it could only be carried out in a converted way: by converting the pressure to the velocity of the wall. When monitoring parameters, such as stress and strain, the stress magnitude of the wall could not be monitored directly; rather,

the stress value was obtained by scanning the combined external force on the wall and dividing it by its contact area. In the model, the time step between each calculation was small enough to make the value of the contact area change negligible.

### 3.2. Thermodynamic Temperature Module

The thermodynamic module of PFC can run independently for simulating the heat transfer of the medium; it can also run coupled with other modules and can be used to analyze the deformation and damage of the medium caused by the interaction of heat and force, which is the application in this simulation.

In this module, the thermodynamic parameters of the different components of the particles needed to be customized, including the specific heat capacity and the coefficient of thermal expansion. According to the tests, the coefficient of thermal expansion mainly controlled the differences in the mechanical properties of the model after the temperature treatment. At the same time, the variation in the specific heat capacity did not significantly influence the mechanical properties curve of the model.

### 3.3. Preparation of Numerical Simulation Model

One of the main elements impacting the calculation time of numerical simulation is the number of model particles. To improve the efficiency of the simulation, the number of model particles should be reduced as much as possible without affecting the simulation results when simulating indoor geotechnical tests. There are two ways to achieve this goal: (1) increasing the particle radius; (2) reducing the specimen size. To enhance the simulation results' reliability, reduction of the number of particles has certain limitations. Richard P. Jessen [32] et al. studied this problem. Their results showed that when the ratio of the specimen diameter D to the average particle diameter d is greater than 30–40, the number of particles will not significantly affect the simulation results. Liu Hong [33] proposed that for sandy soils, of which the particle size is smaller and the difference between the maximum and minimum particle diameter is significant, the number of particles can be reduced by taking the weighted average of the particle radius in the indoor test and enlarging it by a certain number of times. This provides an idea for the reduction of particle numbers in this simulation.

### 3.4. Model Properties

The size of the model in this simulation was set as a standard cylindrical specimen of 50 mm*100 mm. The average tailings particle size according to the measured tailings particle sizes of different gradations processed by the methods mentioned above and guidelines after amplification is shown in Table 1. The dry density is listed in Table 2.

**Table 1.** Average particle diameter of tailings particles (after treatment).

| Group | 1 | 2 | 3 | 4 |
|---|---|---|---|---|
| *d*, mm | 0.632 | 0.761 | 0.931 | 1.140 |

**Table 2.** Dry density of tailings.

| Group | 1 | 2 | 3 | 4 |
|---|---|---|---|---|
| $\rho$, kg/m$^3$ | 1480 | 1640 | 2240 | 2460 |

### 3.5. Parameter Calibration and Model Reliability

The type of modeling was the contact bonding model. The characteristics of this model were that it provided a minimal linear elastic behavior for contact forces, could only carry friction when the contact interface was not bonded, and could not carry friction when the contact interface was bonded. The size of the domain was set to 500 mm * 500 mm, which was sufficient to fully contain the whole model. The initial temperature was set to the usual

room temperature of 25 °C, the temperature increment during freezing was set to −45 °C, and the temperature increment during thawing was set to 40 °C.

The initial reliable model was obtained by repeatedly adjusting each parameter through a trial-and-error method to make the numerical model curve characteristics close to the indoor test curve. The final calibrated partial mechanical parameters are listed in Table 3.

**Table 3.** Calibration results of acceptable view parameters.

| Emod | Fric | Tens | Shears |
|---|---|---|---|
| $1.8 \times 10^8$ | 0.4 | $3.2 \times 10^5$ | $12 \times 10^6$ |

As shown in Figure 1, the initial numerical model curve was close to the laboratory test curve based on the peak stress and strain and the curve change trend, which proved the reliability of the numerical model. The subsequent cyclic freeze–thaw numerical simulations were based on this, and the corresponding results were obtained by adjusting the values of each factor and analyzing them.

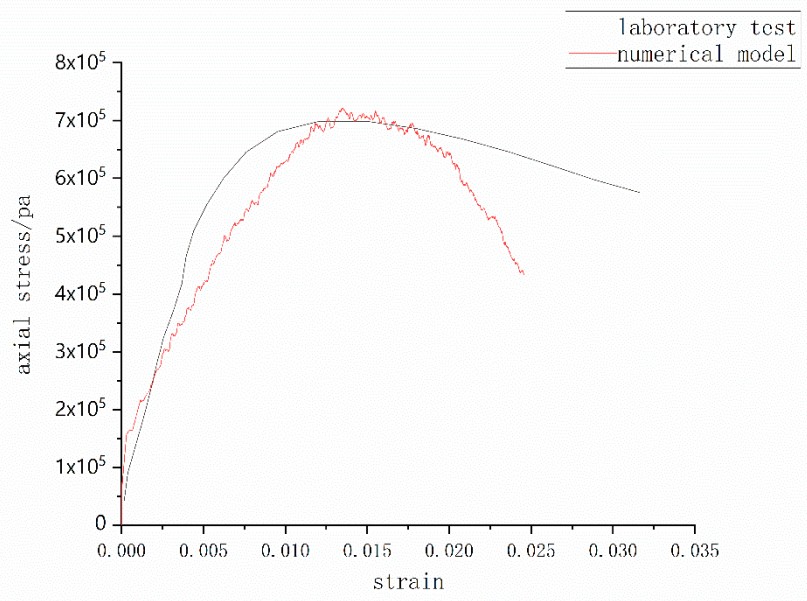

**Figure 1.** Numerical model reliability.

## 4. Analysis of Orthogonal Tests and Mathematical Methods

### 4.1. Orthogonal Test

There were many factors in this simulation. To improve the efficiency of the simulation, the orthogonal test method was used for pre-experimentation to initially investigate the law of three factors, namely the freeze–thaw cycle number, the average particle diameter, and the dry density, on the uniaxial compression strength of tailings. The results were used as the basis for subsequent more in-depth analysis. The results of the orthogonal test are shown in Table 4.

### 4.2. Correlation Analysis

The orthogonal test results were used to examine the correlation between the three independent variables of the freeze–thaw cycle number, dry density, and mean particle diameter on the dependent variable of strength of tailings. The results are shown in Table 5.

**Table 4.** Orthogonal experiment.

| $d$ (mm) | $\rho$ (kg/m$^3$) | Number of Freeze–Thaw Cycles | Uniaxial Compressive Strength (MPa) | Number of Freeze–Thaw Cycles | Uniaxial Compressive Strength (MPa) |
|---|---|---|---|---|---|
| 0.632 | 1480 | 0 | 0.676 | 30 | 0.409 |
| | 1640 | 1 | 0.535 | 24 | 0.410 |
| | 2240 | 3 | 0.531 | 18 | 0.435 |
| | 2460 | 7 | 0.513 | 12 | 0.504 |
| 0.761 | 1640 | 1 | 0.576 | 24 | 0.406 |
| | 2240 | 3 | 0.560 | 18 | 0.457 |
| | 2460 | 7 | 0.522 | 12 | 0.508 |
| | 1480 | 0 | 0.686 | 30 | 0.396 |
| 0.931 | 2240 | 3 | 0.538 | 18 | 0.427 |
| | 2460 | 7 | 0.465 | 12 | 0.456 |
| | 1480 | 0 | 0.633 | 30 | 0.401 |
| | 1640 | 1 | 0.541 | 24 | 0.442 |
| 1.140 | 2460 | 7 | 0.459 | 12 | 0.436 |
| | 1480 | 0 | 0.621 | 30 | 0.405 |
| | 1640 | 1 | 0.497 | 24 | 0.386 |
| | 2240 | 3 | 0.508 | 18 | 0.410 |

**Table 5.** Correlation analysis.

| | Average Particle Size | Dry Density | Number of f-t Cycles | Uniaxial Compressive Strength |
|---|---|---|---|---|
| Average particle size | 1 | 0.000 | 0.000 | −0.196 |
| Dry density | | 1 | −0.185 | −0.127 |
| Number of f-t cycles | | | 1 | −0.846 ** |
| Uniaxial compressive strength | | | | 1 |

** represents at the level of 0.01 (double-tailed), and the correlation between the two factors is significant.

As shown in Table 5, the correlation between the freeze–thaw action number and the uniaxial compressive strength was the most significant and the relationship between the two was negative, which may have been due to the weakening of the fine grain structure of the tailings after repeated freezing and thawing. This does not mean that these two factors had no effect on the strength, but rather reflects the relatively low effect of dry density and mean particle size on the strength of the tailings specimens after the repeated rising and falling of temperature, probably because the density and particle size of the tailings were less affected by temperature in the temperature range of the simulation, and therefore had a lesser effect. As the particle temperature cycled between −25 °C and +25 °C, the final value of the particle temperature did not change, so the volume change in the particles due to temperature was extremely limited, and the effect of the two factors themselves, density and particle size, was also small compared to the cyclic freeze–thaw, and therefore showed a non-significant correlation. The exact pattern of influence and the effect on tailings not subjected to freeze–thaw cycles will be investigated in more detail in the following sections.

*4.3. Regression Analysis*

(1)  Regression parameters

The regression correlation coefficient R$^2$ = 0.838 was calculated by SPSS software, indicating that the fit was acceptable. The results are listed in Table 6.

Since the confidence interval of the dry density of the independent variable contained zero points, this independent variable was deleted, and the regression analysis was carried out with the remaining variables. The results are listed in Table 7.

**Table 6.** Regression coefficient 1.

| | Unnormalized Coefficient | | Significance | Collinearity Diagnostics | | 95% Confidence Interval for B | |
|---|---|---|---|---|---|---|---|
| | B | σ | | Tolerance | VIF | Lower Limit | Upper Limit |
| (constant) | 0.728 | 0.039 | 0.000 | | | 0.658 | 0.817 |
| $d$ | −0.084 | 0.032 | 0.016 | 1.000 | 1.000 | −0.150 | −0.017 |
| $\rho$ | $-4.39 \times 10^{-5}$ | 0.000 | 0.001 | 0.966 | 1.036 | 0.000 | 0.000 |
| F-T cycles | −0.007 | 0.001 | 0.000 | 0.966 | 1.036 | −0.008 | −0.006 |

**Table 7.** Regression coefficient 2.

| | Unnormalized Coefficient | | Significance | Collinearity Diagnostics | | 95% Confidence Interval for B | |
|---|---|---|---|---|---|---|---|
| | B | σ | | Tolerance | VIF | Lower Limit | Upper Limit |
| (constant) | 0.643 | 0.036 | 0.000 | | | 0.569 | 0.716 |
| $d$ | −0.084 | 0.039 | 0.042 | 1.000 | 1.000 | −0.164 | −0.003 |
| F-T cycles | −0.007 | 0.001 | 0.000 | 1.000 | 1.000 | −0.008 | −0.005 |

From Table 7, it can be concluded that the significance of the independent variables was less than 0.05, and the confidence interval did not contain zero points. The VIF was less than 10, indicating no multicollinearity, and the coefficient estimates were reasonable and passed the test. The regression equation was obtained as follows:

$$\hat{y} = 0.643 - 0.084 \times x_1 - 0.007 \times x_2 \tag{1}$$

where $y$ represents uniaxial compressive strength (MPa), $x_1$ represents average particle size (mm), and $x_2$ represents the freeze–thaw cycle number.

(2)    Significance test and residual analysis

The significance test results and residual analysis of the regression model shown in Equation (1) are shown in Table 8 and Figure 2.

**Table 8.** Significance test.

| | Sum of Squares | Degrees of Freedom | Mean Square | F | Significance |
|---|---|---|---|---|---|
| Regression | 0.159 | 2 | 0.080 | 44.460 | 0.000 |
| Residual | 0.052 | 29 | 0.002 | | |
| Aggregate | 0.211 | 31 | | | |

As shown in Table 8, the significance of the regression model was less than 0.05; as shown in Figure 2, the residuals obeyed a normal distribution. Summing up the above two points, it is evident that the model was statistically significant.

From this regression equation, it can be judged that the uniaxial compressive strength of tailings decreases with the average particle size and the increase in the number of freeze–thaw cycles. At the same time, the influence of dry density is relatively small, which further verifies the conclusion in the correlation analysis. The regression equation also shows that the particle size of tailings particles has a greater effect on the strength than the dry density. The effect of particle size on strength may be due to the fact that the larger the average particle size is, the larger the pore volume between the particles, and therefore the lower the macroscopic strength of the model. The subsequent numerical simulations built on the results of the pre-experiments and mathematical analyses to further reveal the relationship between tailings strength and the cyclic freeze–thaws, particle size, and dry density.

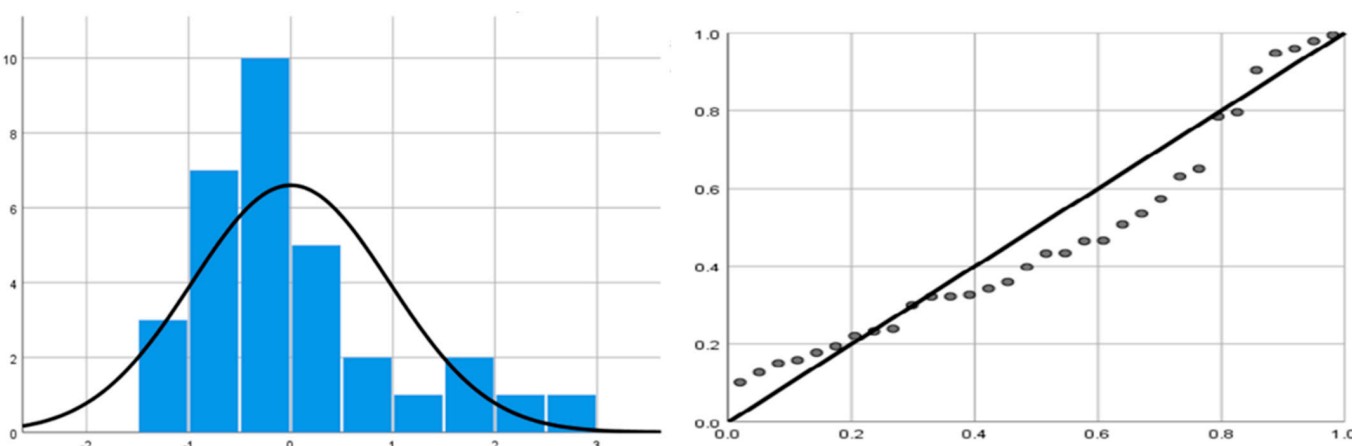

(**a**) Regression standardized residual histogram

(**b**) Regression normalizes the residual standard P-P diagram

**Figure 2.** Residual analysis.

## 5. Numerical Simulation and Analysis of Uniaxial Compression Experiments on Tailings under Freeze–Thaw Cycles

*5.1. Effect of Freeze–Thaw Cycle Number on the Uniaxial Compressive Strength of Tailings*

In this simulation, the freeze–thaw temperature was set to −25 °C versus +25 °C. Some of the stress–strain curves of the model with different dependent variable parameters are shown in Figures 3 and 4.

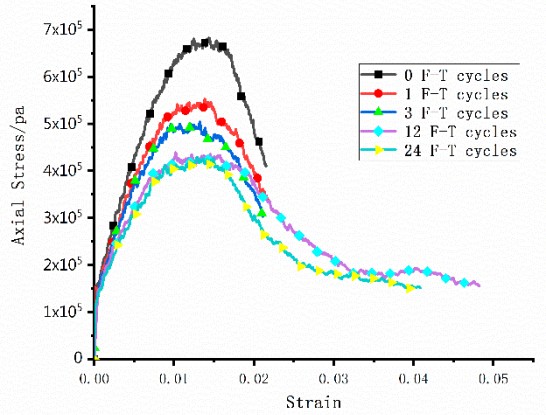

**Figure 3.** Stress–strain curves of tailings samples under different freezing–thawing cycles.

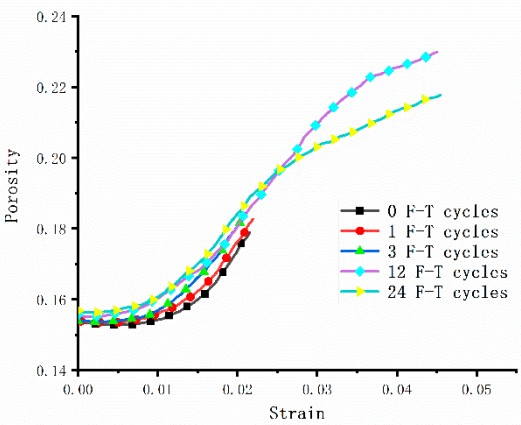

**Figure 4.** Variation in porosity of tailings samples under different freezing–thawing cycle conditions.

It can be seen from Figure 3 that the uniaxial compressive strength of the tailings specimens with the same dry density and average particle size decreased with the freeze–thaw cycle number. The change in the compressive strength of the tailings specimens with the change from 0 to 1 freeze–thaw processes was the largest, with a rate of change of up to 30%; there was also a significant difference in the strength of tailings specimens between three and seven freeze–thaw cycles. After reaching seven cycles, the increase in the freeze–thaw cycle number no longer significantly affected the strength and deformation of tailings specimens, and their stress–strain curves were very similar.

In Figure 4, it can be seen that the porosity of the specimens all showed a tendency to decrease and then increase. The porosity curves of the samples with 0, 1, and 3 freeze–thaw cycles were shorter. In contrast, the porosity curves continued to develop after the number of freeze–thaw cycles reached seven, with a general trend of decreasing slowly, increasing rapidly, increasing slowly, increasing. Part of the reason for the relatively significant differences in stress–strain curves for 0, 1, 3, and 7 freeze–thaw cycles was also due to the considerable differences in porosity. After the number of freeze–thaws reached seven, the difference in the growth phase of the curves was no longer significant, which is also consistent with the fact that the difference in the specimens' stress–strain curves was no longer meaningful. The intercept analysis of the *y*-axis in Figure 4 shows that the samples with more freeze–thaw actions had more significant initial porosity before loading was performed, so it can be assumed that the freeze–thaw cycle leads to an increase in the tailings specimens' porosity, which causes a decrease in the samples' strength on a macroscopic scale. The reason may be that the tailings are a discrete body with a certain amount of air and water in its pores, which constitutes a complex solid–liquid-gas three-phase structure with the solid particles. The change in temperature during the freeze–thaw cycle causes the pore water to freeze and melt repeatedly. Its volume changes periodically, which triggers the expansion and generation of fractures inside the specimen and finally increases porosity.

Figure 5 shows that the cyclic freeze–thaw action did not affect the anisotropic properties of the tailing specimens, and did not affect the main direction of the discrete grouping. The number of contacts of the specimen particles showed a significant change when the number of freezing and thawing reached seven. The number of connections stabilized after seven freeze–thaw cycles and no longer decreased with the number of cyclic freeze–thaw. There was no significant difference in the number of contacts between the specimens after one freeze–thaw process and those after three freeze–thaw actions. However, there was a pronounced difference in the macroscopic uniaxial compressive strength, which indicates that for the tailings specimens after freeze–thaw actions, the number of contacts between the particles was not the main factor affecting the strength. From Figure 6, it can be concluded that uniaxial compression increased the degree of anisotropy of the model subjected to freeze–thawing, but did not change the main direction of the model configuration.

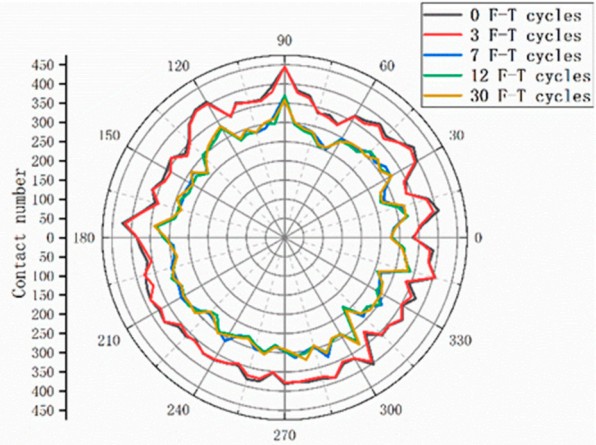

**Figure 5.** Distribution of the number of model contacts for different freeze–thaw cycle numbers.

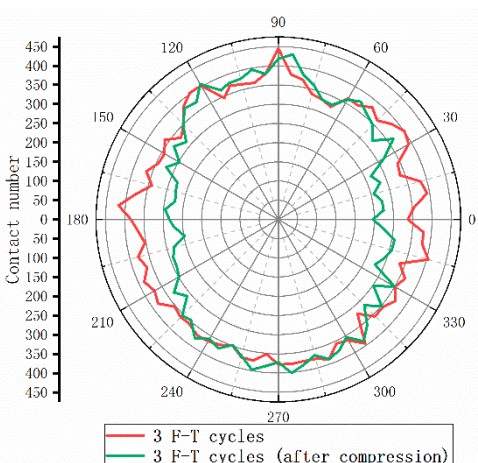

**Figure 6.** Distribution of the number of model contacts before and after uniaxial compression for three freeze–thaw cycles.

As shown in Figure 7, without unfreezing, model sample size distribution and location of the contact force chain were uniform; with the increase in cyclic freeze–thaw, the size and spatial distribution of the contact force chain produced a significant change, and the phenomenon of stress concentration gradually appeared. This suggests that after cyclic freezing and thawing, a test specimen had a stronger capacity for internal fissure formation and expansion. After three freezing–thawing cycles, the distribution of contact force chains remained relatively homogeneous, but there were many obvious stress concentrations inside the model; after seven freezing–thawing cycles, there were obviously more developed cracks inside the model, and the distribution of contact force chains also showed a significant non-uniformity. This indicates that the model was damaged on a macroscopic scale, resulting in the contact force transmission within the model being affected; after the number of freeze–thaw cycles reached 18, the stress concentration phenomenon was more significant and occurred in all parts of the model, indicating that the spatial distribution of the crack expansion within the model was more uniform. It can be deduced that the internal fracture expansion capacity of the tailings after cyclic freeze–thawing had increased, and a large number of stress concentrations and macroscopic fractures had already appeared after seven cycles of freeze–thawing. It can be considered that the damage to the internal structure of the tailings has reached a critical state at this time, and therefore after more than seven freeze–thaw cycles, the strength of tailings does not change significantly with subsequent repeated freezing and thawing.

### 5.2. Effect of Dry Density on the Uniaxial Compressive Strength of Tailings

In this section, model specimens with an average particle diameter of 0.632 mm after three freeze–thaw cycles are used to examine the effect of different dry densities on the strength.

As shown in Figure 8, the uniaxial compressive strength of the tailings specimens with different dry densities had significant differences up to seven cycles of freeze–thaw. When the number of freezing–thawing cycles reached seven, uniaxial compressive strength tended to be stable, which is consistent with the research conclusion of Yonglong Qv [9]. The higher the dry density, the higher the specimens' compressive strength.

As shown in Figures 9 and 10, monitoring the loading process of the tailings model after three cycles of freeze–thaw showed that before loading, the porosity of the models with different dry densities was the same, which indicated that the dry density had no direct influence on the porosity of the specimens. After loading, the larger the dry density of the model, the lower the rate of porosity growth, which indicates that the influence of the dry density on the porosity was mainly reflected in the resistance to pore generation and expansion when the applied load was received. This suggests that the larger the dry

density of the model, the greater its compressive strength after cyclic freeze–thaw. This is similar to Huaqing Wang's proposal that the denser the internal structure of tailings is, the lower the porosity is, which can improve the compressive strength to a certain extent [34].

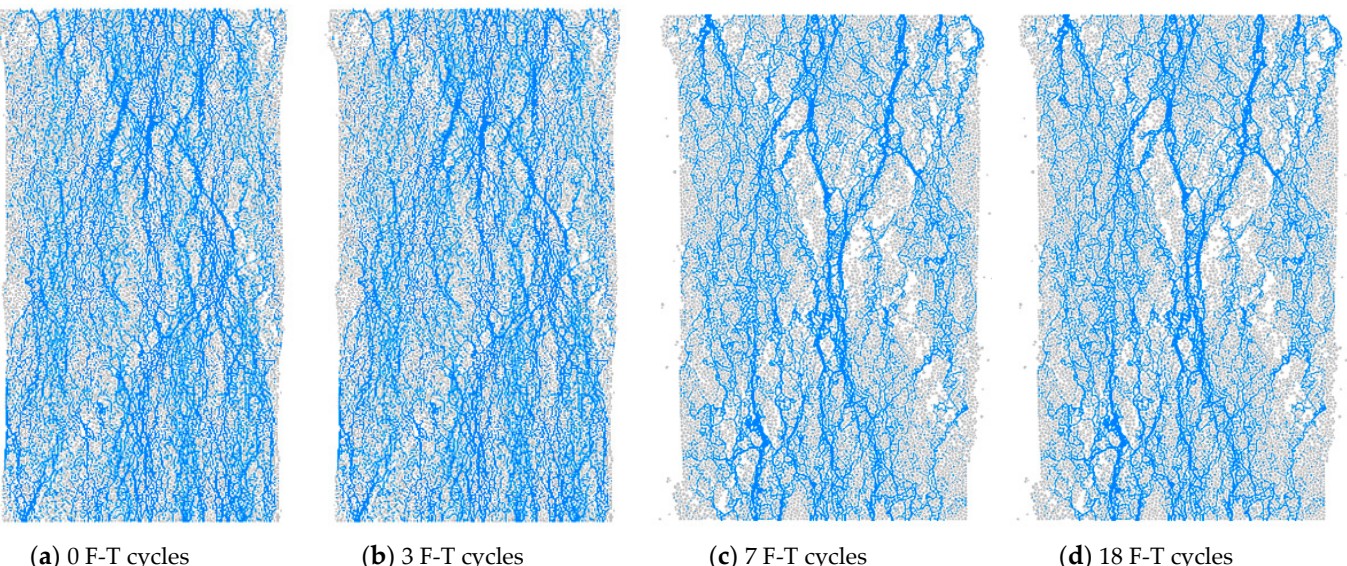

(**a**) 0 F-T cycles　　　　　(**b**) 3 F-T cycles　　　　　(**c**) 7 F-T cycles　　　　　(**d**) 18 F-T cycles

**Figure 7.** Uniaxial compressive particle contact force chains for different freeze–thaw discrete element models.

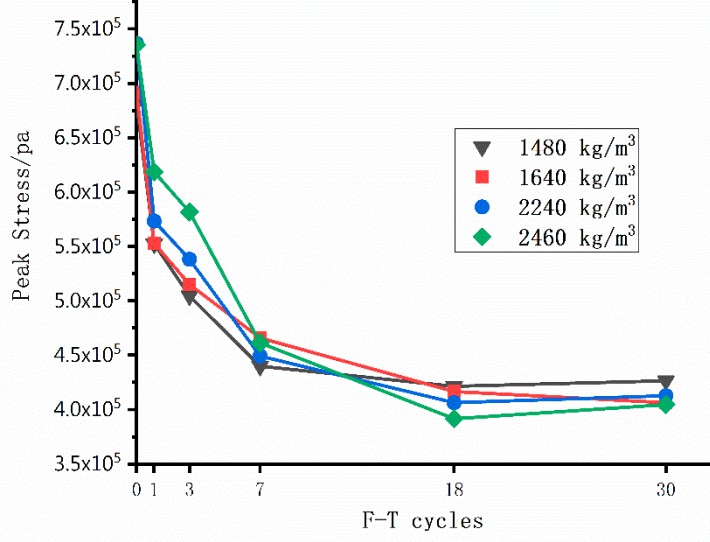

**Figure 8.** Comparison of compressive strength of different dry density tailings model specimens under different numbers of cyclic freeze–thaws.

When the number of freezing–thawing cycles reached seven, the difference in compressive strength of different dry density specimens decreased slightly. In addition, some scholars believe that when the freezing–thawing cycle occurs between 5 and 9 times, the performance of tailings tends to be stable [35].

The relationship between dry density and compressive strength changed. The effect of dry density was no longer significant, consistent with the results analyzed in SPSS software. Some scholars believe that in the freezing–thawing cycle, the quality of tailings samples constantly declines, leading to significant changes in strength [36]; as the freeze–thaw cycle is repeated a certain number of times, the changes in tailings quality and density are no longer obvious, so the influence of dry density on strength is no longer significant. This also shows that the mechanical properties of tailings under freeze–thawing cycles is a complex

mechanical problem. At this time, the factors that directly affect the strength of the sample are cohesion and internal friction angle. Some scholars found that these two factors tended to be stable only after the number of freezing–thawing cycles reached nine [37].

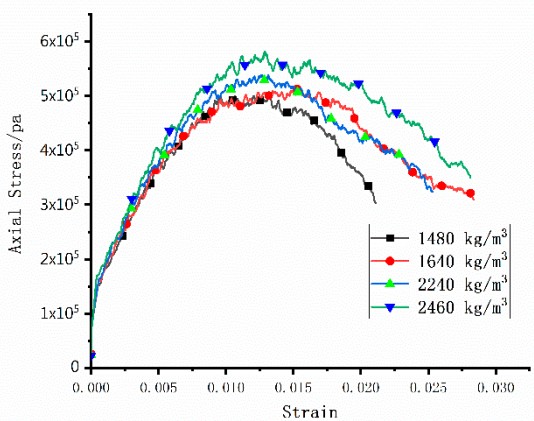

**Figure 9.** Stress–strain curves of specimens with different dry densities under three freeze–thaw actions.

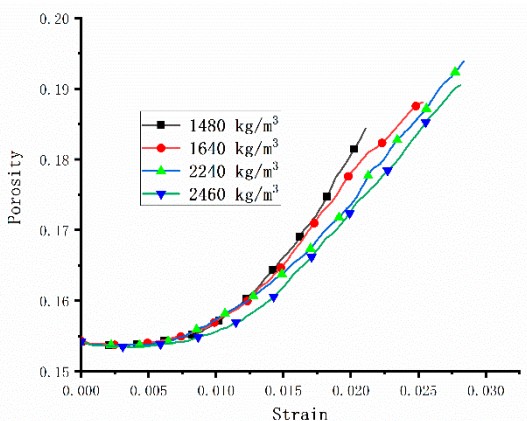

**Figure 10.** Porosity changes of specimens with different dry densities under three freeze–thaw actions.

It was known that the particle contact number distribution of the tailings model after three freeze–thaw cycles was not affected by the cyclic freezing and thawing, so the contact numbers of the model specimens before undergoing three freeze–thaw cycles were analyzed and investigated in this section. The results showed that the distribution of contact quantities for models with different dry densities had some differences. However, the values of the amounts and the anisotropy characteristics were generally close to each other, as shown in Figure 11.

Figure 12 shows that the location distribution of contact force chains had an apparent relationship with the spatial distribution of fissures, and macroscopic cracks were easily generated on structural surfaces with significant differences in the strength of contact force chains. Therefore, tailings models with more uniform spatial distribution as well as the power of contact force chains have relatively higher strength. It can be seen that dry density had no obvious influence on the distribution and size of the contact force chain of the model. Therefore, considering comprehensively, the influence of dry density on strength was mainly reflected in the ability of the model to resist pore formation and development when subjected to external load.

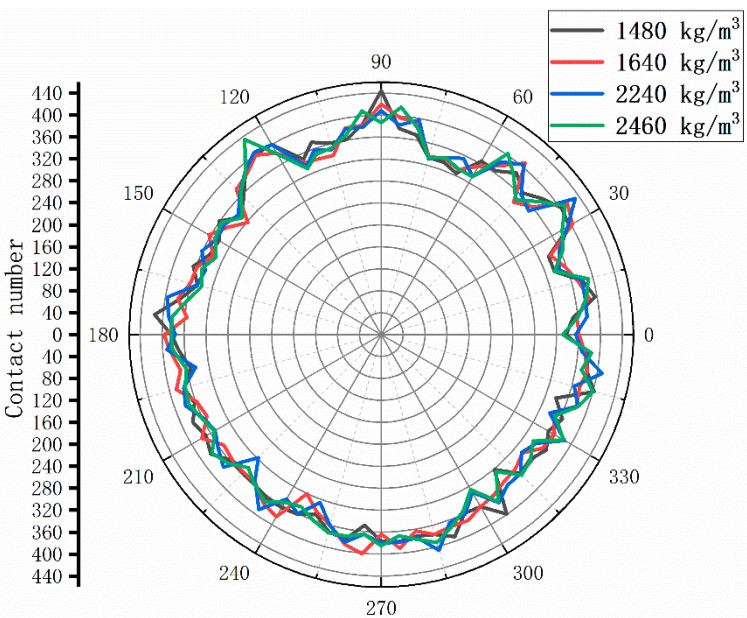

**Figure 11.** Distribution of the number of contacts of different dry density models after three cyclic freeze–thaws.

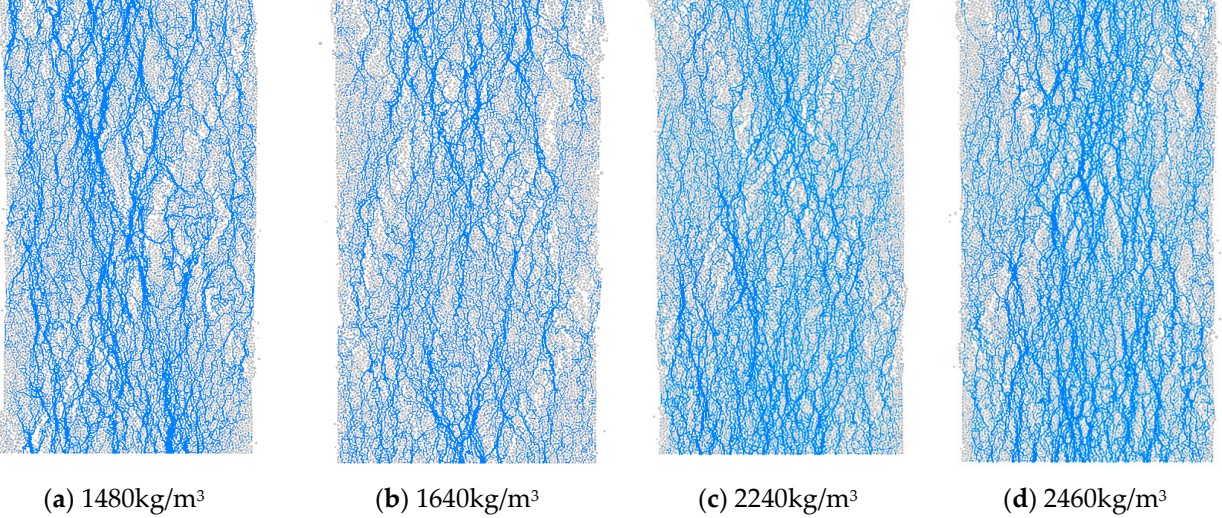

(**a**) 1480kg/m³     (**b**) 1640kg/m³     (**c**) 2240kg/m³     (**d**) 2460kg/m³

**Figure 12.** Comparison of contact force chains of specimens with different dry density models (three occasions of cyclic freezing–thawing).

### 5.3. Effect of Average Particle Size on the Tailings' Uniaxial Compressive Strength

In this section, tailings with a dry density of 1480 kg/m$^3$ were used as the research object to investigate the effect of average particle diameter on specimens on strength with different numbers of freeze–thaw cycles, and the results are shown in Figure 13.

Figure 13 shows that the relationship between the average particle size and compressive strength conforms to the rule that the larger the average particle diameter of the tailings model, the lower its compressive strength. Comparing Figures 14 and 15, it is evident that the model compressive strength was directly related to its porosity size, and the larger the particle size, the larger the pores between the particles, resulting in a lower macroscopic compressive strength.

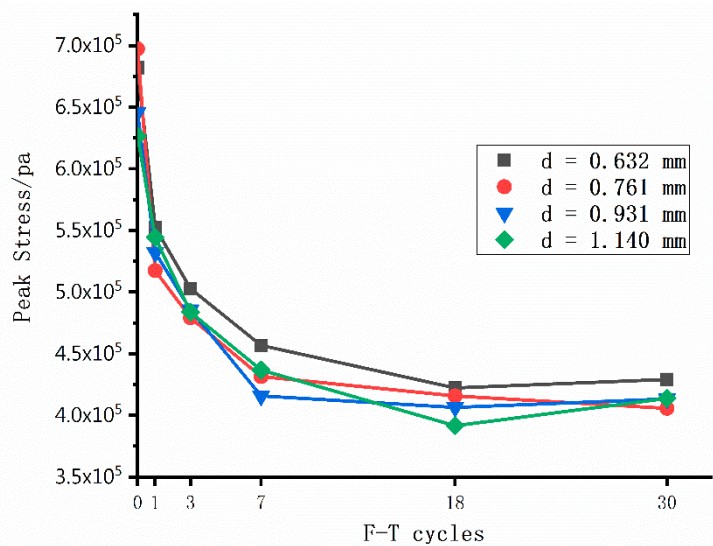

**Figure 13.** Comparison of compressive strength of models with different numbers of freezing and thawing cycles.

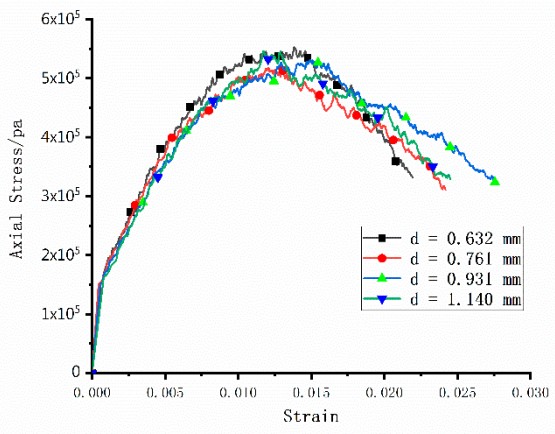

**Figure 14.** Stress–strain curves of specimens with different particle sizes with one freeze–thaw cycle.

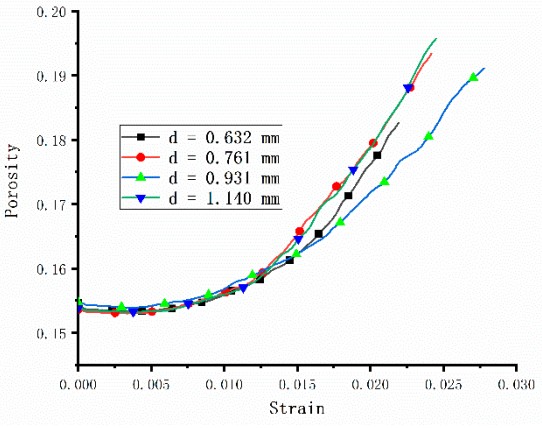

**Figure 15.** Variation in porosity of specimens with different particle sizes with one freeze–thaw cycle.

## 6. Conclusions

Through the analysis of the test results by orthogonal test and SPSS software, it is evident that the influence of the number of freezing–thawing cycles and the average particle diameter on the uniaxial compressive strength was relatively significant. In contrast, the effect of the dry density was relatively insignificant.

(1) The regression model of the tailings' uniaxial compressive strength on the average particle diameter and the number of freeze–thaw cycles was obtained, with significant statistical significance and reliability.

(2) With the increase in the number of freeze–thaw cycles, the compressive strength of tailings decreased from 0.686MPa to 0.4MPa. After 30 F-T cycles, the strength dropped by 40%. The more significant the average particle diameter, the lower the compressive strength of tailings; when the particle size increased by 80%, the strength decreased by only 8%, which was even smaller after the freeze–thaw cycle; and after regression analysis, the higher the dry density, the higher the compressive strength of tailings. There was a significant decrease in the number of contacts in the tailings specimens when the number of freeze–thaw cycles reached seven, but it remained relatively stable before and after seven freeze–thaws. It is thus inferred that the number of contacts was not the main factor affecting the compressive strength of the tailings specimens.

(3) The analysis of the porosity variation curve shows that the porosity of the specimen increased with the number of freeze–thaw cycles, and directly affected the sample's macroscopic compressive strength. The average particle size directly affected the porosity, while the dry density affected the porosity by resisting the generation and expansion of pores when subjected to the applied load.

**Author Contributions:** G.H.: methodology, project administration, conceptualization; Y.Y.: numerical simulation, methodology, formal analysis, writing—original draft; M.L.: visualization, writing—review and editing; J.Z.: investigation, validation, data curation; F.L.: supervision; A.D.M.: revision of manuscript grammar; H.Y.: funding, resources. All authors have read and agreed to the published version of the manuscript.

**Funding:** This research was funded by [National Key R&D Project of China] grant number [No. 2020YFC1909602 and Grant No. 2021YFC2902901] and [Key R&D Project of Hubei province] grant number [No. 2021BCA152].

**Conflicts of Interest:** The authors declare there is no conflict of interest.

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
