# Peer review of "Numerical Study on Mechanical Properties of the Freezing–Thawing Cycle of Tailings Based on Particle Discrete Element Method"

_minerals, doi:10.3390/min12070904_

Round 1
Reviewer 1 Report
The thesis starts from the potential safety problems of tailings ponds in alpine regions, and uses a combination of numerical simulation and theoretical analysis to study the influence law of three factors, namely the number of freeze-thaw cycles, the particle size and dry density of tailings, on the performance of tailings materials, and elaborates on the causes and mechanisms. The paper has a clear research idea, complete structure, sufficient content and reliability, and has certain reference value for the study of safety and stability of tailings pond sub-dams in alpine regions, and is recommended to be accepted after revision.
The following suggestions are put forward for revision:
1. It is suggested to simplify the contents of Table 4, and the current table 4 is somewhat jumbled;
2. Recommends that the content of the conclusions in part 5 be streamlined, with consideration given to reducing the number of articles in the conclusions;
3. How is the number of freeze-thaw cycles determined in the simulation?
4. There is an author's name in the introduction with irregularities in case.
Author Response
Response to Reviewer 1 Comments
Point 1: It is suggested to simplify the contents of Table 4, and the current table 4 is somewhat jumbled.
Response 1: Thanks for your suggestion. We redrew the Table 4 to make it more clear.
Point 2: Recommends that the content of the conclusions in part 5 be streamlined, with consideration given to reducing the number of articles in the conclusions.
Response 2: Thanks for your suggestion. We have streamlined the conclusion section and reduced the number of entries appropriately.
Point 3: How is the number of freeze-thaw cycles determined in the simulation?
Response 3: Thanks for your questions. According to the actual situation of the mine in an alpine area, and considering the influence of a larger number of freezing-thawing cycles on tailings dam, the current one is selected.
Point 4: There is an author's name in the introduction with irregularities in case.
Response 4: Thanks for your suggestion. We've made changes and checked the rest.

Reviewer 2 Report
General comments:
The manuscript entitled Numerical study on mechanical properties of the freezing-thawing cycle of tailings based on particle discrete element method presents some interesting numerical results on the main factors affecting the mechanical behaviour of tailings after freezing and drying. The authors used the PFC2D to assess the effects of F-T cycles, dry density, and particle-size diameter on tailings’ properties. Overall, the authors present several interesting results. However, because the methodology in not well presented and provides only limited details, it is hard to appreciate the quality and relevance of the results. Therefore, I recommend the paper to be rejected, but reconsidered after some major revisions. The authors will find below a list of specific comments / suggestions to help improve the quality of their manuscript. Since I believe that improving sections 1 and 2 will significantly improve the overall quality of the manuscript, I oriented my specific comments towards those two sections. I also strongly suggest the authors to have their manuscript edited by a science-specific editor.
Specific comments:
Introduction :
Line 42-44 : this statement is not clear and does not refer to any relevant literature. The disposal method usually refers to the types of tailings (i.e. conventional / pulp, thickened or filtered tailings) and the construction method (i.e. upstream, downstream or central axis). Most of the problems associated with tailings disposal are with conventional tailings disposed in tailings storage facilities constructed with the upstream method.
Line 45-46: The authors should refer to permafrost (permafrost is a material that is at or below 0°C for at least two consecutive years) and seasonally frozen ground. Perennial and seasonal permafrost are not usual definitions associated with frozen ground. This very important and should be clear and modified throughout the manuscript.
Line 48-51: This is a far-fetched statement and both references are associated with the freeze-thaw behaviour of intact rocks. The structural strength and stability could be affected by F/T cycles, but this still has to be demonstrated …
Line 52-54: add reference.
Line 54-56: As the introduction is written, I don’t see the research need. I suggest the authors to articulate their introduction around the influence of F/T cycles on the properties of the tailings (how F/T cycles affect the strength of tailings) and how this could be paramount to the stability of tailings dams. It should also be notes that F/T cycles are strictly a surface phenomenon [active layer for permafrost regions and seasonal F/T for other cold regions]. Therefore, the emphasis should be on how this surface layer could induce large ground movement.
Line 57-63: Multiple ideas in the same sentence, please revise.
Line 62-87: There is a lot of information dropped in that section, but I have difficulty finding the general stream. The authors list several factors affecting the behaviour of many types of materials while they should have focussed on tailings and other tailings-like materials such as silts and silty-clays.
Line 62 et throughout the manuscript: Use of indoor tests is inappropriate. I suggest using laboratory tests or tests performed at controlled temperatures, etc.
Line 130 to 141: The objectives of the article are not clear. I suggest the authors to state the objectives clearly. The first objective is … The second objective is … This will the reader better understand what is said in those 10 lines. I also suggest adding a sentence highlighting the originality of the work with respect to the literature. From what we get in the introduction, it is a little hard to place the contribution of this study within what is already available. What is the gap that this study is filling. The introduction should be rewritten and shortened to fulfill this comment.
Section 2:
I suggest adding a section presenting the general methodological approach. As it is now, it starts abruptly, and the readers don’t have an overall idea of the methodology before going into the specifics.
The authors should consider adding / modifying several sections.
(1) Materials properties are given at lines 187-190 and 194-195 and this seems out of place. Making a materials properties section could help with that.
(2) Laboratory testing: Section 2.4 presents results from laboratory tests used to calibrate the model. However, the authors did not mention that any laboratory tests were performed. This should be said, and the test parameters should be given.
(3) PFC2D model: Section 2.1: presents that PFC2D is divided into three main parts: (1) the model generation part; (2) the thermal treatment part; and (3) the loading part. It would be easier to present the model as such. It should also be clearly stated what the model is designed for. What are the heat transfer mechanisms that the model can solve? Does it solve conduction and phase change? Then, what about stress modeling? How is F/T modelled? This is important.
(4) Model configuration and approach: The authors do not explain at all their modelling strategy (i.e. kind of modelling matrix). The manuscript should provide model configuration and parameters for each of the results presented in section 3. i.e. I believe that section 2.3 should be named Numerical simulation parameters (or something similar - there is no such thing like ‘specimens’ in numerical simulations) and detail the above. Questions like what is the domain size, mesh, time steps, initial conditions (for stress and temperature), what the materials’ thermal properties and water content, boundary conditions, 1D, 2D, 3D? … This is also very important.
Other specific comments:
Line 170-171: Numerical simulations can not replace geotechnical testing. Numerical modelling can be used to obtain an estimate of a given behaviour, asses some parameters of influence or extend trends / behaviours observed in the field or at the lab, but never replace laboratory testing. This section should be modified accordingly.
Line 173 – 175: This is called model optimization. Add reference.
Line 176: not sure increasing the particle radius is an effective way to achieve this goal since it changes the materials properties. If so, the authors should add some key reference.
Line 177 to 185: this should be in the introduction.
Author Response
Response to Reviewer 2 Comments
Point 1: Line 42-44: this statement is not clear and does not refer to any relevant literature. The disposal method usually refers to the types of tailings (i.e. conventional / pulp, thickened or filtered tailings) and the construction method (i.e. upstream, downstream or central axis). Most of the problems associated with tailings disposal are with conventional tailings disposed in tailings storage facilities constructed with the upstream method.
Response 1: Thanks for your suggestion. The methods dealing with tailings here means to storage tailings or use tailings, for example, as some kind of building material. This may be due to improper use of words or to differences in the meaning of words in English and Chinese contexts.
Now we get the point and have modified this statement: Many scholars have studied the problems in tailings disposal, at present, the most important form of tailings storage facilities in the world is still tailings dam.
Point 2: Line 45-46: The authors should refer to permafrost (permafrost is a material that is at or below 0°C for at least two consecutive years) and seasonally frozen ground. Perennial and seasonal permafrost are not usual definitions associated with frozen ground. This very important and should be clear and modified throughout the manuscript.
Response 2: Thanks for pointing out this problem. We have revised the entire manuscript in the parts that deal with these two concepts.
Point 3: Line 48-51: This is a far-fetched statement and both references are associated with the freeze-thaw behaviour of intact rocks. The structural strength and stability could be affected by F/T cycles, but this still has to be demonstrated.
Response 3: Thanks for pointing out this problem. We initially believe that rock is the building material for the initial stage of tailings dam, so its characteristics can reflect the stability of tailings dam to a certain extent. Now we have modified this statement and referred to a reference on the study of tailings dams in freezing-thawing zone.
Point 4: Line 52-54: add reference.
Response 4: Thanks for your suggestion. We’ve added reference here.
Point 5: Line 54-56: As the introduction is written, I don’t see the research need. I suggest the authors to articulate their introduction around the influence of F/T cycles on the properties of the tailings (how F/T cycles affect the strength of tailings) and how this could be paramount to the stability of tailings dams. It should also be notes that F/T cycles are strictly a surface phenomenon [active layer for permafrost regions and seasonal F/T for other cold regions]. Therefore, the emphasis should be on how this surface layer could induce large ground movement.
Response 5: Thanks for your suggestion. It is by means of numerical simulation that we investigated the effect of freeze-thaw cycles on the tailings properties, and proved the feasibility of PFC numerical simulation software applied to tailing materials. And considering that permafrost in alpine regions often has a certain depth, it is of practical importance to study the effect of freeze-thaw cycles on the dam body of tailings dams. The research in this paper is just the beginning of a series of studies and an important foundation for how changes in tailings properties affect the overall stability of tailings dams, which is still being studied.
Point 6: Line 57-63: Multiple ideas in the same sentence, please revise.
Response 6: Thanks for pointing out this problem. We have revised this sentence.
Point 7: Line 62-87: There is a lot of information dropped in that section, but I have difficulty finding the general stream. The authors list several factors affecting the behaviour of many types of materials while they should have focussed on tailings and other tailings-like materials such as silts and silty-clays.
Response 7: Thanks for pointing out this problem. We deleted the part about rock, cement mixture and slope stability, and kept the part about clay. In addition, studies on temperature and cycle times in freezing-thawing cycles are very important and provide valuable references for research, so these parts are also retained.
Point 8: Line 62 et throughout the manuscript: Use of indoor tests is inappropriate. I suggest using laboratory tests or tests performed at controlled temperatures, etc.
Response 8: Thanks for your suggestion. We have revised the relevant parts of the manuscript.
Point 9: Line 130 to 141: The objectives of the article are not clear. I suggest the authors to state the objectives clearly. The first objective is … The second objective is … This will the reader better understand what is said in those 10 lines. I also suggest adding a sentence highlighting the originality of the work with respect to the literature. From what we get in the introduction, it is a little hard to place the contribution of this study within what is already available. What is the gap that this study is filling. The introduction should be rewritten and shortened to fulfill this comment.
Response 9: Thanks for your suggestion. We have rewritten and shortened this section and clearly highlighted the objectives of the article.
Point 10: I suggest adding a section presenting the general methodological approach. As it is now, it starts abruptly, and the readers don’t have an overall idea of the methodology before going into the specifics.
Response 10: Thanks for your suggestion. We added a methodological part after the introduction to briefly explain the research methods.
Point 11: Materials properties are given at lines 187-190 and 194-195 and this seems out of place. Making a materials properties section could help with that.
Response 11: Thanks for your suggestion. We have added a material properties section to show the properties.
Point 12: Laboratory testing: Section 2.4 presents results from laboratory tests used to calibrate the model. However, the authors did not mention that any laboratory tests were performed. This should be said, and the test parameters should be given.
Response 12: Thanks for your suggestion. In fact, that laboratory test investigated the effect of the ratio of cement to tailings on the strength of the mixed cementitious mass, and only the test results of the pure tailings in it were used to calibrate the numerical model in the study of this paper, and the number of relevant parameters of the experiment was also small, so that the experiment was not mentioned in this paper.
Point 13: PFC2D model: Section 2.1: presents that PFC2D is divided into three main parts: (1) the model generation part; (2) the thermal treatment part; and (3) the loading part. It would be easier to present the model as such. It should also be clearly stated what the model is designed for. What are the heat transfer mechanisms that the model can solve? Does it solve conduction and phase change? Then, what about stress modeling? How is F/T modelled? This is important.
Response 13: Thanks for your suggestion and questions. We have further explained the purpose for which these parts are designed.
The heat transfer mechanism of the model is to define thermodynamic parameters for all particles in the model, including heat transfer coefficient, thermal expansion coefficient, etc. Then the particles in the model (some or all) are assigned temperature parameters, and then the model will perform calculations according to the assigned temperature and thermodynamic parameters. The temperature distribution and other states of the model will constantly change according to the operation process, so the model can solve the heat transfer problem. However, since the particle size is fixed, the model cannot directly simulate the phase transformation process. However, the model can indirectly represent the change of model porosity and other parameters during the phase transformation process by changing the contact relationship and contact type between particles.
In the loading section, the walls are generated at the top and bottom of the model and a velocity is applied to the walls, and the model is loaded in this way.
The freeze-thaw cycle is implemented in the heat treatment part by assigning an initial temperature to the model and controlling the temperature increments to achieve the freeze-thaw cycle.
Point 14: Model configuration and approach: The authors do not explain at all their modelling strategy (i.e. kind of modelling matrix). The manuscript should provide model configuration and parameters for each of the results presented in section 3. i.e. I believe that section 2.3 should be named Numerical simulation parameters (or something similar - there is no such thing like ‘specimens’ in numerical simulations) and detail the above. Questions like what is the domain size, mesh, time steps, initial conditions (for stress and temperature), what the materials’ thermal properties and water content, boundary conditions, 1D, 2D, 3D? … This is also very important.
Response 14: Thanks for your suggestion. We have added some details of the modeling in section 2.4 (section 3.5 after adding the section used to introduce the methodology)
Point 15: Line 170-171: Numerical simulations can not replace geotechnical testing. Numerical modelling can be used to obtain an estimate of a given behaviour, asses some parameters of influence or extend trends / behaviours observed in the field or at the lab, but never replace laboratory testing. This section should be modified accordingly.
Response 15: Thanks for pointing out this problem. We have modified this section accordingly.
Point 16&17: Line 173 – 175: This is called model optimization. Add reference.
Response 16&17: Thanks for your suggestions. In fact, this is a problem caused by our improper paragraphing of the article. After describing two approaches to reduce the number of models, the next paragraph presents the study conducted by Jenson and Liu Hong et al. to address this aspect. We have adjusted the paragraphs in this section.
Point 18: Line 177 to 185: this should be in the introduction.
Response 18: Thanks for your suggestion. As we mentioned in response 16&17, this paragraph is intended to provide a discussion of the feasibility of reducing the number of particles in numerical simulations. We had some errors in the line segmentation of this section, and we have now made the changes.
The authors really appreciate the reviewer’s suggestions for our manuscript and pointed out the defects of our paper.

Reviewer 3 Report
-abstract. shorten, and report the most important results found
-page 13. add some references to discuss the results
-introduction-Separate paragraphs that contain more than 10 lines
-Please, describe and inform the contact model of the PFC2D
-Conclusions. Point out the conclusions with numerical results
Author Response
Response to Reviewer 3 Comments
Point 1: abstract. shorten, and report the most important results found.
Response 1: Thanks for your suggestion. We have streamlined the abstract, and reported most important results found.
Point 2: add some references to discuss the results.
Response 2: Thanks for your suggestion. Some references are added and discussed in combination with the results in page 13.
Point 3: Separate paragraphs that contain more than 10 lines.
Response 3: Thanks for your suggestion. We have adjusted the introduction accordingly.
Point 4: Please, describe and inform the contact model of the PFC2D.
Response 4: Thanks for your question.
The contact bonding model is used for numerical simulation. The characteristics of this model are that it provides a minimal linear elastic behavior for contact forces, and can only carry friction when the contact interface is not bonded, and cannot carry friction when the contact interface is bonded (as show in FIG.1 and FIG.2). Contact interfaces cannot resist relative rotation and must be in either a bonded or unbonded state. When the contact interface is bonded, the contact shape is always linear elastic until the bond is broken after the strength limit is exceeded.
FIG.1 Contact bonding model
FIG.2 Bond broken
Point 5: Conclusions. Point out the conclusions with numerical results.
Response 5: Thanks for your suggestion. We added some numerical results to the conclusion to improve it.

Reviewer 4 Report
Title: Numerical study on mechanical properties of the freezing-thawing cycle of tailings based on particle discrete element method
Comments:
The work is focused on the mechanical properties of tailings after exposure to freeze-thaw cycles. The manuscript is presented well. However following changes are required before accepting the manuscript.
Line 16 & 18: Write full forms of PFC and SPSS software.
Line 27: “The overall…..increasing” is a confusing statement. Please elaborate on the limitation.
The novelty of the present research must be defined in the Abstract and Introduction.
In the Introduction section, The effect of freeze-thaw cycles can be elaborated for Biocemented soils using the following references:
1. Sharma M, Satyam N, Reddy KR (2021) Effect of freeze-thaw cycles on engineering properties of biocemented sand under different treatment conditions. Eng Geol 284:106022. https://doi.org/10.1016/j.enggeo.2021.106022
2. Sharma M, Satyam N, Reddy KR (2022) Liquefaction Resistance of Biotreated Sand Before and After Exposing to Weathering Conditions. Indian Geotech J 52:328–340. https://doi.org/10.1007/s40098-021-00576-x
Flow charts can be used to describe the methodology.
The manuscript must be revised for punctuation, grammatical errors, spacing
Line 178: correct the referencing style.
Figure 3, Figure 4, Figure 9, Figure 10, Figure 14, and Figure 15 should be revised, the curves should be provided with symbols so that difference can be optimized in a black and white printout also. X-Y axis titles must be bold for clear visibility.
In conclusion, first summarize the work, then write about specific findings
Author Response
Response to Reviewer 4 Comments
Point 1: Line 16 & 18: Write full forms of PFC and SPSS software.
Response 1: Thanks for your suggestion. The full form of PFC is Particle Flow Code, the full form of SPSS is Statistical Product Service Solutions. And we've added them to the article.
Point 2: Line 27: “The overall…..increasing” is a confusing statement. Please elaborate on the limitation.
Response 2: Thanks for your suggestion. Actually, this sentence is a misnomer in our Chinese writing, which leads to this kind of misrepresentation in our English translation. And what the original sentence meant was: “The change of porosity generally shows a decreasing - slowly increasing - smoothly increasing trend”.
In fact, two other reviewers suggested that we simplify the abstract, so we removed the sentence after consideration.
Point 3: The novelty of the present research must be defined in the Abstract and Introduction.
Response 3: Thanks for your suggestion. We made some adjustments to the summary to show novelty.
Point 4: In the Introduction section, The effect of freeze-thaw cycles can be elaborated for Biocemented soils using the following references:
Response 4: Thanks for your suggestion. We have carefully read these two references, which are obviously of high research value, but they are mainly aimed at the study of biological cements, which is not relevant to the study of tailings in this paper. However, these two articles have some enlightening effect on us: this biological effect may be used to reinforce tailings dams. Therefore, if there are related researches in the future, these two references can play their due value more.
Point 5: Flow charts can be used to describe the methodology.
Response 5: Thanks for your suggestion. We have drawn flow charts before, but considering that the method of this study is not complicated, we did not add flow charts in the manuscript. The flow chart is shown below:
FIG.1 flow chart
Point 6: The manuscript must be revised for punctuation, grammatical errors, spacing.
Response 6: Thanks for your suggestion. We checked and corrected these errors in the article. Dr. Akisa David Mwangi in our research team came from English-speaking countries and the grammar of the article.
Point 7: Line 178: correct the referencing style.
Response 7: Thanks for your suggestion. We have corrected that referencing style.
Point 8: Figure 3, Figure 4, Figure 9, Figure 10, Figure 14, and Figure 15 should be revised, the curves should be provided with symbols so that difference can be optimized in a black and white printout also. X-Y axis titles must be bold for clear visibility.
Response 8: Thanks for your suggestion. We have adjusted these figures accordingly. We have adjusted these graphs accordingly to ensure that they are distinguishable even in black and white print, and X-Y axis has been bold for clear visibility.
Point 9: In conclusion, first summarize the work, then write about specific findings.
Response 9: Thanks for your suggestion. We revised the conclusions and separated out the summary of the conclusion and the findings.

Round 2
Reviewer 4 Report
The manuscript can be accepted in its current form as all the comments and suggestions are incorporated.